# Strength and Power Characteristics in National Amateur Rugby Players

**DOI:** 10.3390/ijerph18115615

**Published:** 2021-05-24

**Authors:** Diego Alexandre Alonso-Aubin, Moisés Picón-Martínez, Iván Chulvi-Medrano

**Affiliations:** 1Wingsport Management SL, 28850 Madrid, Spain; moi_beneixama@hotmail.com; 2UIRFIDE (Sport Performance and Physical Fitness Research Group), Department of Physical and Sports Education, University of Valencia, 46010 Valencia, Spain; ivan.chulvi@uv.es

**Keywords:** squat, bench press, training, strength, speed

## Abstract

Rugby players need muscular strength and power to meet the demands of the sport; therefore, a proper assessment of the performance in rugby players should include both variables. The purpose of this study was to examine the strength and power characteristics (SPC) during the squat (SQ) and bench press (BP) in national amateur rugby players and to analyze gender- and position-related differences. A total of 47 players (30 males and 17 females; age: 25.56 ± 1.14 and 23.16 ± 1.38 years, respectively) participated in the study. The one repetition-maximum (1-RM) and SPC in SQ and BP were obtained using a Smith Machine. Then, subjects performed one set of five repetitions on the SQ and BP against six relative loads (30–40–50–60–70–80% 1-RM) using a linear transducer. Differences between genders were found in 1-RM for maximal power, kilograms lifted at maximal power, maximal power, maximal strength and maximal speed in BP (*p* < 0.00) and 1-RM, kilograms lifted at maximal power, maximal power, maximal strength and maximal speed in SQ (*p* < 0.00). Comparisons between variables in SQ and BP present a significant relationship (*p* < 0.01) in SQ and BP 1-RM with kilograms lifted at maximal power *(r* = 0.86 and r = 0.84), maximal strength (*r* = 0.53 and *r* = 0.92) and maximal power (*r* = 0.76 and *r* = 0.93). This study confirms the importance of the SPC assessment for training prescription in rugby amateur players.

## 1. Introduction

Rugby is a collision sport that involves high-intensity bouts of exercise including sprint and agility activities and contact and tackling separated by short bouts of low-intensity activity [1,2]. Rugby players need speed, agility, muscular strength and power to meet the demands on the sport and these factors distinguish high- and low-level players [3,4]. Indeed, muscular strength and power are directly associated with performance, whereby the elite players demonstrate the highest muscle power values [5,6]. For example, high-speed running demands are influenced by strength and power and, consequently, by the force–power–velocity profile (FVP) characterizing the maximal mechanical capabilities of the neuromuscular system [7]. Moreover, due to the tactical and movement patterns of rugby, players should have agility skills for avoiding contact and collisions [8,9,10,11]. In the case of rugby, there are two general player positions, backs and forwards, with different physical demands [12,13]. Forwards are involved in more collisions, whereas backs are involved in more high-speed running (>5 m·s^−1^) [14]. In addition, previous studies identified that forwards are stronger and more powerful than backs [15]. Backs are reportedly faster and more agile than forwards [16,17].

Muscular strength and power are key attributes for rugby players due to the contact and collision element of the sport, alongside its relationship with those for other physical qualities [18]. Maximal strength is the vehicle that drives the development of the strength and power and allows athletes to develop greater performance on speed and agility activities [19,20]. Thereby, when considering the ability to develop power, it is clear that high levels of muscular strength are a key factor to reach high levels of power [21,22].

Assessing the physical demands could assist in athlete development, guiding athletes’ training and assisting coaches. For example, training studies that incorporate maximal or explosive strength exercises have found improvements in the sprinting speed of athletes [21]. In addition, there are several studies presenting strength and power data via Wattbike peak power output, countermovement jump or isoinertial highlighting the importance on strength and power characteristics (SPC) understanding [23].

However, studies on SPC in females are limited, which causes a gap in the knowledge about SPC in females and if there are differences in comparison to males. We hypothesize that males are stronger and more powerful than females because they are heavier and bigger [24,25]. It seems that at the elite level in females, forwards were heavier and displayed greater upper-body strength, whereas backs showed greater acceleration and maximal speed abilities [26]. Recently, similar data were reported in other studies comparing backs and forwards where high-speed demands were different, suggesting that maximal velocity running and strength and power training are important [27]. These facts highlight the importance of assessing SPC and maximizing long-term adaptation of muscle force and power via resistance training [28]. Consequently, a proper assessment of the performance in rugby players should include maximal strength and power. However, there is a gap in the research on the study of these two capacities and their relationship for rugby performance and how strength and conditioning programs should be individualized attending to position demands and gender differences. In order to assess SPC, exercises should be selected that provide a transfer to the sport in skill movement and strength. Thus, the squat (SQ) and the bench press (BP) are two of the most used and effective exercises in resistance training for strengthening the lower and the upper body for improving athletic performance [29,30].

Therefore, the purpose of this study was to examine the strength and power characteristics (SPC) on the SQ and BP in national amateur rugby players and to analyze sex- and position-related differences for better strength and conditioning program designs. We hypothesize that there are differences in SPC between backs and forwards and males and females.

## 2. Materials and Methods

### 2.1. Participants

A total of 47 rugby players, with at least 5 years of experience in practice in rugby and resistance training performing parallel SQ and BP exercises, volunteered to participate in this study. Exclusion criteria included a musculoskeletal injury over the past six months, any medical condition that could limit the exercise performance, and taking steroids, drugs, medications or dietary supplements for enhancing sport performance. Subjects were national amateur rugby players and included 30 males (age 26.56 ± 1.14 years; height 1.78 ± 0.16 m; and body mass 86.85 ± 1.88 kg) and 17 females (age 23.16 ± 1.38 years; height 1.63 ± 0.16 m; and body mass 66.46 ± 2.39 kg). Then, for comparing player positions we split in forwards (age 25.57 ± 1.25 years; height 1.74 ± 0.15 m; and body mass 84.11 ± 2.43 kg; 20 males and 10 females) and backs (age 25.07 ± 1.27 years; height 1.71 ± 0.25 m; and body mass 72.23 ± 3.04 kg; 10 males and 7 females). Their average weekly training volume was 13 h*wk^−1^ including three days of rugby-specific training, three days of resistance training and competition. All procedures followed ethical principles for medical research involving human subjects by the World Medical Association Declaration of Helsinki (General Assembly of the World Medical Association. 2014). The study was approved by the University of Alicante Institution Ethics Committee UA-2018-06-20. All subjects were informed of the purpose of the study, and experimental procedures and potential risks of the study. They were given the opportunity to ask any questions related to the procedures. After being informed, they signed the informed consent form.

### 2.2. Procedures

The present research was performed in 3 different sessions, all separated by 3 days. The first visit to the laboratory was dedicated to informing the subjects about the procedures, familiarization with tests and anthropometric measurements. Maximum strength tests (1-RM) in parallel SQ and parallel BP were performed in the second session and power assessment in the third in randomized order. For the strength assessments, we used a Smith Machine (Multipower, Line Selection; Technogym, Gambettola, Italy). All the sessions were performed at the same time, between 5 p.m. and 8 p.m. All tests were supervised by a Certified Strength and Conditioning Specialist (CSCS) who has 10 years’ experience in testing and training rugby athletes. Subjects were instructed to avoid any strenuous physical activity 24 h prior to each assessment and not to eat and drink water ad libitum 45 min before the assessments. Before testing, players performed a standardized 15 min warm-up that consisted of 5 min of pedaling on a cycle ergometer at an intensity of 50 watts followed by 1 set of 15 repetitions on either SQ or BP exercise at an increasing velocity with a 20-kg barbell.

#### 2.2.1. Squat and Bench Press Techniques

To standardize exercise performance during the testing sessions, an analogic goniometer was used to measure 90° of knee flexion (parallel-squat) and ensure a consistent stance distance during the SQ and 90° of elbow flexion and biacromial distance of the grip width during the BP.

To ensure the correct range of motion, straps were used to limit a greater displacement than 90° of knee flexion on the SQ and 90° of elbow flexion on the BP.

During the SQ, the load was lifted without lifting heels off the ground, keeping the back straight, eyes focused forward and feet slightly wider than shoulder-width apart with toes pointing slightly outward.

During the BP, the load was lifted without lifting the hips off the bench, with the neck and the back lying on the bench and the feet on the ground [31,32].

#### 2.2.2. Maximal Strength

Subjects completed the squat assessment first and then the bench press assessment. Subjects performed 1 set of 3–4 repetitions on the SQ and BP exercises with 4 relative loads calculated according to their previous 1-RM performed 2 weeks ago in a training-test session (60–70–80–90% 1-RM) for warming up. After the specific 15 min warm-up previously mentioned, subjects performed a 1-RM attempt by increasing progressively (by 10–20% in the SQ and 5–10% in the BP) the load used in 100% 1-RM. If the subject failed the 1-RM attempt, we decreased the load by 5–10% in the squat and 2.5–5% in the BP. The subjects’ 1-RMs were achieved within five attempts. Subjects were given 3 min of recovery between each set and 5 min between exercises.

#### 2.2.3. Power Strength

After 72 h from the maximal strength assessment, subjects returned to the testing facility to perform 1 set of 5 repetitions on the SQ and BP exercises on 6 relative loads with an increasing intensity calculated from the data recorded in the first session (30–40–50–60–70–80% 1-RM). Subjects were given a 3 min recovery between each set and 5 min between exercises.

Participants were told the importance of performing the concentric phase at the highest speed and effort possible. During the performance, they were not given any kind of feedback.

#### 2.2.4. Measurement Equipment and Data Analysis

The bar was properly instrumented with a linear position transducer (T-Force System, Ergotech, Murcia, Spain) that has a precision in 1000 N and a sampling frequency on 1000 Hz for maximal power recording. This device has been used to assess kinetic and kinematic variables in resistance exercises. The system consists of a linear velocity transducer extension cable in interface with a personal computer that obtains data with an analogic–digital resolution of 14 bits. The specific software (TFDMS Version 2.35) calculates the kinematic and kinetic parameters of each repetition, and stores and provides all information from the results obtained in real time [33]. The system’s software automatically calculated the bar velocity of every repetition, providing auditory and visual feedback in the same moment of realization.

The concentric phase or positive work was as fast as the subject could perform. The eccentric or negative work, and recovery phase had a duration of 3.5 s [34]. Additionally, all the measurement data were stored on a virtual disk.

Subsequently, the software analyzed the data, obtaining the following variables for both exercises: maximal power at a given percentage (Max. Power at 1-RM%), kilograms used to achieve the highest power value (Max. Power kg), maximal power (Max. Power in W), maximal strength (Max. Strength in N), maximal speed (Max. Speed), time spent reaching maximal power (Time to Max. Power), time spent reaching maximal speed (Time to Max. Speed).

### 2.3. Statistical Analysis

The normality of the data for each group was checked using the Shapiro–Wilk test. Due to the normal distribution, data are described as mean and standard deviation (SD). One-way ANOVA was used to determine differences between backs and forwards, and men and females. Pearson correlations were performed to determine the significance of the association between variables (0.00 to 0.30: negligible correlation; 0.30 to 0.50: low positive correlation, 0.50 to 0.70: moderate positive correlation, 0.70 to 0.90: high positive correlation, 0.90 to 1.00: very high positive correlation) [35]. Significance was set at *p* < 0.05. To assess effect size d, the Cohen test was used. The effect size indices were 0.2 = small; 0.5 = medium; 0.8 = large and 1.3 = very large [36]. Analyses were performed using SPSS^®^ v25.0 for Mac (SPSS, Inc., Chicago, IL, USA).

## 3. Results

Table 1 shows differences by gender and rugby position in 1-RM for both exercises, maximal power at a given percentage (Max. Power at 1-RM%), kilograms used to achieve the highest power value (Max. Power kg), maximal power (Max. Power in W), maximal strength (Max. Strength in N), maximal speed (Max. Speed), time spent reaching maximal power (Time to Max. Power), time spent reaching maximal speed (Time to Max. Speed).

Significant differences (*p* < 0.01) were found between males and females in the BP in 1-RM (d = 14.67), kilograms used to achieve the highest power value (d = 14.77), maximal power (d = 1.84), maximal strength (d = 1.26) and maximal speed (d = 4.20) and in the SQ in kilograms used to achieve the highest power value (d = 2.96), maximal power (d = 0.50) and maximal speed (d = 3.99).

Table 2 and Table 3 show the comparisons between positions by gender among variables in SQ and BP. We found significant differences (*p* < 0.01) in the BP in females in kilograms to achieve the higher power values (d = 0.76) and maximal strength (d = 0.81).

Table 4 and Table 5 show the correlations among variables in SQ and BP.

## 4. Discussion

The purpose of this study was to examine SPC on the SQ and BP in national amateur rugby players, analyzing possible differences in performance between genders and playing positions. One finding of this study was that the assessment of the SPC could be a useful tool for resistance-training prescription, adding the ability for coaches to prescribe training according the role demands. In addition, our study could provide the opportunity for other studies to recruit samples from multiple clubs, thus increasing sample sizes, generalizability of results and statistical power of subcategory comparisons (e.g., position, playing level or gender). This objective is one of the goals proposed in the most recent scientific literature [37].

First, there is a scarcity in the literature comparing SPC in males and females, and we found strong differences between BP and SQ in males and females, in both absolute and relative values of the different variables of the SPC, especially in 1-RM, maximal power and maximal speed in SQ and BP. These results are similar to those found in previous research conducted with adolescent rugby players where there were also differences between genders in both exercises [38,39]. These findings could be attributed to the prevalence of slower type-I and II-A fibers in females compared with males that parallels the lower contractile velocity in females compared with males and differences in thyroid hormone, estrogen and testosterone levels [40]. Supporting this, one study previously demonstrated that with different maturation status SPC could be different in males and females, as in adolescent age females could demonstrate greater power values than males [38].

Secondly, comparing different sports disciplines also shows that the variables that make up the SPC are specific not only to sports but also individually [41]. In the case of rugby, the player’s role may have an important implication for the training design since the specific techniques of the sport as well as the movements associated with their playing positions may have a direct relationship with the training method to be applied [42]. Supporting this evidence, other findings confirm the influence of training and sport activity on the force and velocity capacity balance for power-oriented sports [43]. Due to these differences and the scarce studies carried on rugby it is important to determine the SPC in these players. One of the objectives of the SPC studies is to determine if there are differences between athletes based on their role during the game. We did not find strong differences between backs and forwards in our study in the SPC variables, even when we compared positions by gender, excepting in maximal speed in the BP exercise when we compared all the subjects together, and maximal power and strength in females. This could be explained by their amateur level, as other studies found differences in strength and maximal force production in BP in rugby union players because of the physical demands of these respective positions [6]. In the literature there are few studies that determine differences related to the athletes’ roles [44,45]. The SPC assessment could be important to determine the factors that have an influence on performance individually on rugby players. Our data show that SQ and BP present a significant strong relationship in 1-RM and maximal power and BP and kilograms lifted at maximal power, maximal strength and maximal power. The sport position could have an influence in the performance showing differences between forwards and backs in maximal speed in BP exercise.

Our study found high correlations in SQ and BP between maximal power and maximal strength, and kilograms lifted at maximal power, suggesting that strength performance is critical for greater power values. Similar conclusions were reported in previous studies in SQ [46,47,48] and in BP [49,50]. Knowing the SPC of the players can also be decisive to know the level and potential of future performance of the players since the strength and power values are considered indicators of the level of development [51]. We analyzed the data and the relationships that exist between the variables of the SPC in order to determine which are the most significant in power performance to prescribe more specific training according to the objectives and for enhancing players’ performance by their roles. Based on our data, maximal power in SQ and BP is strongly related to maximal strength. In this sense, the training contents can be manipulated by the coaches to make rugby players improve their performance through increased production of strength [52].

In the same way, the relationship between the force–velocity profile (FVP) and performance in sprint and jump tests has also been studied, determining that there is a high correlation between some of the FVP variables such as maximum power, speed and strength, especially in the first one [53].

Therefore, it is necessary to develop new research to understand different training protocols in order to improve the performance of the FVP associated with the specific context of rugby, including also the assessment of speed and agility since it will allow determining the possibilities of improvement in movement during game situations. In the case of trying to improve rugby performance, it is suggested that resistance training must be adjusted using SPC values close to the rugby role playing reported in high performance players. This will optimize the SPC in the different positions in rugby forwards or backs.

This is the first study analyzing the SPC in national amateur rugby players by comparing the SPC variables between gender and positions and providing SPC data in females by positions. However, some limitations should be acknowledged. Research in SPC in national amateur rugby players is a field with gaps in the literature and we need more studies for a better understanding. In our study, we only had access to 47 rugby players, but we know that a greater number of subjects would be better for improved inferences. In this sense, we have not assessed the muscle mass, and this could give a better perspective about the SPC and its relationship with the body composition. Finally, we have not included a rugby-specific test to correlate with the SPC.

## 5. Conclusions

This study confirms the importance of the SPC assessment for training prescription in national amateur rugby players for enhancing performance by player position or gender. In addition, this study provides data so that other investigations can compare their results and thus establish a database to make inferences about the performance of rugby players. It is confirmed that there are differences between males and females in both absolute and relative values of the different variables of the SPC, especially in 1-RM, maximal power and maximal speed in SQ and BP. These variables are also critical for sport performance and should be considered for a proper assessment and training in rugby performance.

## Figures and Tables

**Table 1 ijerph-18-05615-t001:** Differences between genders and positions.

	Males	Females	Significance	Effect Size	Forwards	Backs	Significance	Effect Size
	Mean ± SD	Mean ± SD	*p*	d	Mean ± SD	Mean ± SD	*p*	d
1-RM Bench Press (kg)	96.83 ± 3.40	37.81 ± 2.13	0.001 **	14.67	76.37 ± 5.99	76.17 ± 8.21	0.89	0.01
Max. Power BP (1-RM%)	61.33 ± 2.47	69.37 ± 2.49	0.02 *	−2.61	62.41 ± 2.74	67.05 ± 2.05	0.30	−1.58
Max. Power BP (kg)	58.93 ± 2.66	25.87 ± 1.37	0.001 **	14.77	46.27 ± 3.88	49.41 ± 4.56	0.53	−0.35
Max. Power BP (W)	713.28 ± 27.84	241.51 ± 15.90	0.001 **	1.84	560.96 ± 51.34	529.11 ± 57.71	0.80	0.02
Max.Strength BP (N)	777.38 ± 27.16	343.64 ± 25.26	0.001 **	1.26	613.69 ± 48.68	652.17 ± 54.16	0.51	−0.03
Max.Speed BP (m/s)	1.06 ± 0.3	0.78 ± 0.42	0.001 **	4.20	1.02 ± 0.04	0.86 ± 0.05	0.04 *	0.10
Time to Max. Power BP (ms)	512.36 ± 23.84	522 ± 42.06	0.82	−0.02	518.13 ± 25.34	499.17 ± 38.33	0.55	0.04
Time to Max. Speed BP (ms)	555.26 ± 22.85	560 ± 44.35	0.90	−0.01	561.41 ± 24.09	536.88 ± 51.09	0.47	0.03
1-RM Squat (kg)	199.83 ± 6.97	125.31 ± 8.23	0.001 **	2.56	170 ± 10.34	180.58 ± 10.23	0.41	−0.20
Max. Power SQ (1-RM%)	68.33 ± 1.92	66.87 ± 3.50	0.85	0.37	67.58 ± 2.30	68.23 ± 2.60	0.94	−0.22
Max. Power SQ (kg)	136.66 ± 6.08	82.00 ± 6.07	0.001 **	2.96	114.58 ± 8.09	122.88 ± 8.19	0.50	−0.25
Max. Power SQ (W)	1472.94 ± 66.57	656.60 ± 45.75	0.001 **	0.50	1154.19 ± 92.54	1248.39 ± 125.19	0.48	−0.02
Max.Strength SQ (N)	1784.16 ± 71.47	1366.42 ± 242.53	0.04 *	0.03	1569.60 ± 106.77	1794.44 ± 207.45	0.23	−0.02
Max.Speed SQ (m/s)	0.95 ± 0.31	0.67 ± 0.43	0.001 **	3.99	0.85 ± 0.04	0.84 ± 0.05	0.78	9.76
Time to Max. Power SQ (ms)	626 ± 37.31	514.23 ± 46.64	0.07	0.13	598.34 ± 39.26	574.11 ± 49.19	0.76	0.02
Time to Max. Speed SQ (ms)	668.50 ± 37.56	572.11 ± 47.14	0.12	0.11	650.03 ± 38.79	615.11 ± 49.69	0.64	0.04

BP: Bench Press; SQ: Squat; s: seconds; 1-RM: one maximum repetition; kg: kilograms; W: watts; N: newtons; ms: milliseconds; * *p* < 0.05; ** *p* < 0.01; d: d Cohen.

**Table 2 ijerph-18-05615-t002:** Squat comparisons between positions by gender.

	Gender	Position	Mean ± SD	Significance (*p*)	Effect Size (d)
1-RM Squat (kg)	Males	Forwards	195.25 ± 44.7	0.362	0.14
		Backs	209 ± 18.52		
	Females	Forwards	113.88 ± 31.2	0.11	0.34
		Backs	140 ± 31.09		
Max. Power SQ (1-RM%)	Males	Forwards	68.5 ± 10.89	0.9	0.05
		Backs	68 ± 10.32		
	Females	Forwards	65.55 ± 15.89	0.68	0.06
		Backs	68.57 ± 12.14		
Max. Power SQ (kg)	Males	Forwards	133.6 ± 36.91	0.48	0.1
		Backs	142.8 ± 25.24		
	Females	Forwards	72.33 ± 22.48	0.69	0.45
		Backs	94.42 ± 21.95		
Max. Power SQ (W)	Males	Forwards	1404.15 ± 383.06	0.14	0.3
		Backs	1610.54 ± 295.24		
	Females	Forwards	598.72 ± 132.39	0.15	0.28
		Backs	731.03 ± 221.09		
Max.Strength SQ (N)	Males	Forwards	1722.93 ± 417.43	0.232	0.21
		Backs	1906.62 ± 317.81		
	Females	Forwards	1228.86 ± 744.22	0.44	0.11
		Backs	1634.19 ± 1322.35		
Max.Speed SQ (m/s)	Males	Forwards	0.93 ± 0.18	0.14	0.3
		Backs	0.98 ± 0.13		
	Females	Forwards	0.69 ± 0.18	0.51	0.09
		Backs	0.63 ± 0.15		
Time to Max. Power SQ (ms)	Males	Forwards	616.2 ± 218.95	0.69	0.67
		Backs	648 ± 180.57		
	Females	Forwards	558.66 ± 200.2	0.38	0.13
		Backs	468.57 ± 196.73		
Time to Max. Speed SQ (ms)	Males	Forwards	656.9 ± 219.81	0.67	0.07
		Backs	691.7 ± 183.11		
	Females	Forwards	634.77 ± 193.98	0.2	0.23
		Backs	460.71 ± 189.43		

BP: Bench Press; s: seconds; 1-RM: one maximum repetition; kg: kilograms; W: watts; N: newtons; ms: milliseconds; d Cohen.

**Table 3 ijerph-18-05615-t003:** Bench Press comparisons between positions by gender.

	Gender	Position	Mean ± SD	Significance (*p*)	Effect Size (d)
1-RM Bench Press (kg)	Males	Forwards	95 ± 17.84	0.456	0.113
		Backs	100.50 ± 20.60		
	Females	Forwards	35 ± 9.68	0.14	0.3
		Backs	41.42 ± 5.56		
Max. Power BP (1-RM%)	Males	Forwards	60 ± 15.55	0.457	0.113
		Backs	64 ± 8.43		
	Females	Forwards	67.77 ± 12.01	0.48	0.1
		Backs	71.42 ± 6.90		
Max. Power BP (kg)	Males	Forwards	56.75 ± 16.23	0.253	0.204
		Backs	63.30 ± 9.77		
	Females	Forwards	23 ± 4.78	0.01 *	0.76
		Backs	29.57 ± 4.64		
Max. Power BP (W)	Males	Forwards	716.99 ± 169.97	0.855	0.054
		Backs	705.88 ± 117.85		
	Females	Forwards	214.23 ± 53.89	0.04 *	0.52
		Backs	276.58 ± 60.83		
Max.Strength BP (N)	Males	Forwards	759.36 ± 167.47	0.357	0.148
		Backs	813.44 ± 99.85		
	Females	Forwards	289.98 ± 55.03	0.008 **	0.817
		Backs	421.79 ± 113.02		
Max.Speed BP (m/s)	Males	Forwards	1.10 ± 0.23	0.196	0.248
		Backs	0.99 ± 0.17		
	Females	Forwards	0.85 ± 0.14	0.07	0.42
		Backs	0.69 ± 0.19		
Time to Max. Power BP (ms)	Males	Forwards	493.45 ± 126.67	0.269	0.193
		Backs	550.2 ± 136.49		
	Females	Forwards	573 ± 148.95	0.08	0.04
		Backs	426.28 ± 167.47		
Time to Max. Speed BP (ms)	Males	Forwards	537.8 ± 117.21	0.288	0.192
		Backs	590.2 ± 139.48		
	Females	Forwards	613.88 ± 147.65	0.09	0.04
		Backs	460.71 ± 189.43		

BP: Bench Press; s: seconds; 1-RM: one maximum repetition; kg: kilograms; W: watts; N: newtons; ms: milliseconds; * *p* < 0.05; ** *p* < 0.01; d: d Cohen.

**Table 4 ijerph-18-05615-t004:** Correlation among squat variables.

	1-RM Squat (kg)	Max. Power SQ (kg)	Max. Power SQ (1-RM%)	Max. Strength SQ (N)	Max. Power SQ (W)	Max. Speed SQ (m/s)
1-RM Squat (kg)						
Max. Power SQ (kg)	0.867 **					
Max. Power SQ (1-RM%)	−0.082	0.436 **				
Max.Strength SQ (N)	0.530 **	0.634 **	0.332 *			
Max. Power SQ (W)	0.764 **	0.808 **	0.228	0.533 **		
Max.Speed SQ (m/s)	0.269	0.165	−0.186	−0.183	0.628 **	
Time to Max. Power SQ (ms)	0.262	0.399 **	0.304 *	0.105	0.367 *	0.22
Time to Max. Speed SQ (ms)	0.208	0.361 *	0.344 *	0.152	0.344 *	0.175

SQ: Squat; s: seconds; 1-RM: one maximum repetition; kg: kilograms; W: watts; N: newtons; ms: milliseconds; * *p* < 0.05; ** *p* < 0.01.

**Table 5 ijerph-18-05615-t005:** Correlation among bench press variables.

	1-RM Bench Press (kg)	Max. Power BP (kg)	Max. Power BP (1-RM%)	Max. Strength BP (N)	Max. Power BP (W)	Max. Speed BP (m/s)
1-RM Bench Press (kg)						
Max. Power BP (kg)	0.843 **					
Max. Power BP (1-RM%)	−0.444 **	0.078				
Max.Strength BP (N)	0.921 **	0.956 **	−0.124			
Max. Power BP (W)	0.938 **	0.821 **	−0.363 *	0.921 **		
Max.Speed BP (m/s)	0.539 **	0.15	−0.747 **	0.303 *	0.621 **	
Time to Max. Power BP (ms)	−0.158	0.168	0.607 **	−0.081	−0.213	
Time to Max. Speed BP (ms)	−0.138	0.161	0.568 **	−0.083	−0.2	−0.302 *

BP: Bench Press; s: seconds; 1-RM: one maximum repetition; kg: kilograms; W: watts; N: newtons; ms: milliseconds; * *p* < 0.05; ** *p* < 0.01.

## Data Availability

Data available on request due to restrictions, e.g., privacy or ethical.

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
