# Peer review of "Strength and Power Characteristics in National Amateur Rugby Players"

_ijerph, 2021, doi:10.3390/ijerph18115615_

Round 1
Reviewer 1 Report
Dear authors, I know that at this point a large amount of work has been done and expectations are high. Because of this, I give credit to all manuscripts I revise, trying to contribute to an upgraded version of the submitted file. So, all the questions I pose are willing to make authors reflect and to address the issues raised, correct them or explain them more deeply.
I have to make a disclaimer: I'm not authoring or co-authoring any of the literature I address in the comments.
Abstract
Please check all the questions below and then revise the abstract in accordance.
Introduction
Authors provide adequate context for their topis and cite other relevant research. I think the specific readership will find this topic meaningful. Nevertheless, despite stating the purpose of the manuscript, I could not find in the introduction any info related to or confirming differences in strength and power variables of backs and forwards. When authors pose the research question (L-53 to L-55), readers cannot find if the better Strength and conditioning programs will be due to the gender differences (hypothesize that there will be gender differences [I think all of us would bet there would]), or to differences in PSC between backs and forwards (hypothesize there will be differences between backs and forwards [this might be a good research question that your data could support and follow your pretension towards the conclusion (L-22, abstract; L233, conclusions)]). Check Duthie et al. (2003), https://doi.org/10.2165/00007256-200333130-00003.
If authors think this suggestion is useful, please organize the intro section to expose information about the differences between backs and forwards. Please check if the following literature helps:
Agar-Newman et al. (2017). https://doi.org/10.1519/JSC.0000000000001167
Brown et al. (2014a). https://doi.org/10.1123/ijspp.2013-0129
Brown et al. (2014b). https://doi.org/10.1123/ijspp.2013-0129
de Lacey et al. (2014). https://doi.org/10.1519/JSC.0000000000000397
Harty et al. (2019). https://doi.org/10.1519/JSC.0000000000003314
Methods
The only question I pose here is whether mixing males with females when considering backs and forward groups created some bias in the results and further interpretation as males are 15 cm taller on average and 20kg heavier than females?
Readers could also find it useful to have the standard warm-up description (Silva et al. (2018). https://doi.org/10.1007/s40279-018-0958-5).
All procedures are well described and can be replicated by other researchers.
Statistics seem proper.
Results
Again, here I raise this question, were there differences in the group’s characteristics? For sure, there were. Then, could any results be different if the results were normalized to body mass?
As a reader, I cannot interpret the forward/back results as I know they could be biased by group composition (forwards 20 males/10 females; backs 10 males/7 females).
The data of Tables 2 and 3 could be presented only in the table. It becomes redundant and is more visual with the table only.
L-140 and 148 – If it is about correlations authors are addressing; please do not refer to comparisons as this might confuse readers.
Again, I am not sure about the relevance of some correlations as some variables are a direct function of others (e.g., P = F x v). Why didn’t authors run the correlations separated by gender?
Comment
The authors begin the discussion with the objective description. Then authors make a statement that is a little ambiguous, speculative, and is not supported to any extent by data (L-165 – “the main finding…”). I call attention here to the heterogeneity, which might be due to the sampling in the players’ position. In L-168 continues and repeats the same idea as before.
The sentence in L-169 to L-171 is hard to read and understand. It seems that there was a correlation between SQ and BP “SQ and BP present significant strong relationship in 1-RM and BP…”; well, authors, better than anyone, know what they meant here.
The next sentence, L-171, presents a very risky statement. I recall my concerns here. If there are very large differences between males and females, how can we interpret the differences between backs and forwards when the groups have males and females together?
The following paragraphs are disconnected from each other, and the findings are not deeply discussed but only corroborated.
Author should focus more on the results and less on the results' own pretension of individualizing /training/enhancing performance/training protocols as data, and research design does not permit to infer any of these questions.
The authors pointed the limitations, and some of them are related to the simple study design. It is worth noting that several studies already published and cited have similar sample sizes or even lower.
Authors do not present practical applications drawn from data.
Conclusion
The conclusion is succinct, but I do not think the data clearly justifies the conclusions
References
Are specific and up to date.
Author Response
Thank you very much for reviewing the manuscript in order to improve its quality.
Without a doubt, we have taken into consideration all your annotations and we hope that the work carried out will be to your liking.
However, we remain open for any consideration you may wish to make.

Reviewer 2 Report
This study aimed to compare between sexes and playing positions the different mechanical outputs obtained during the back-squat and bench press exercises in national amateur rugby players. In general, the authors should make a considerable effort to highlight the novelty of the study, since it is topic widely examined in literature, as well as they should clarify some major methodological issues that have not been addressed in the manuscript. It should be noted that part of the current literature has not been contemplated in the manuscript. In addition, although the paper is readable, the writing needs a lot of work before the manuscript is selected for publication. Please see below for specific comments or queries
Abstract
Line 9. Check the grammar of "requires".
Line 11. Change "in" by "during".
Line 11. What does "power-strength characteristics" mean?
Line 14. Explain the abbreviation for "1RM" the first time it is introduced in the text.
Line 15. Change “on” by “against”.
Lines 16-21. I'm sorry, but I can't understand the results: 1) Wouldn't maximum strength be the 1-RM?, 2) maximum power and maximum velocity at what load ?, and 3) Are you referring to comparisons or associations? Ultimately, the writing of the results has to improve considerably.
Introduction
Sorry, but the introduction is very brief and some unstructured. I recommend that the authors 1) contextualize rugby, 2) expose the importance of power and strength in this sport, as well as the testing protocol used in the literature, 3) state if there are differences based on sex, and 4) state if there are differences depending on the position of the players. In other words, the authors should present the body of literature with respect to the objectives and highlight what is the novelty of the present study.
Line 53-55. What are the hypotheses? Again, the authors should expose the body of literature that supports the expected results.
Material and methods
Lines 80-82. Do you mean by "maximum strength tests" the determination of 1-RM? Please be more explicit, the maximum strength can be determined in different ways. What does "power strength" mean?
Lines 93-94. How is consistency ensured during squat and bench press exercises?
Lines 93-97. Describe the exercises in more detail. For example, was the downward movement performed in a controlled manner? Does the barbell contact the chest during the bench press? What execution technique was followed during the bench press exercise? It would be recommended that this information be presented in another subsection.
Line 98. How did you decide if it was 3 or 4 repetitions?
Line 104. Which exercise is performed first?
Lines 105-109. Authors should present the "Measurement equipment and data analysis" in another subsection and provide more detail on how the different variables are obtained.
Lines 111-116. In what order are the external loads applied? Is any kind of feedback provided? Was the maximum value or the average value used for data analysis?
Lines 121-122. What scale was used to interpret Pearson’s correlation coefficients?
Lines 122-123. Check the grammar of this sentence.
Results
Again, I recommend that the authors explain in detail how the different variables were calculated. For example, mean force is discouraged as a valid variable when comparing force output using linear position transducers.
https://pubmed.ncbi.nlm.nih.gov/29923856/
Table 2. P-values can never be zero. Therefore, express them as P <0.001.
For a better understanding, I recommend that the authors report descriptive values in one figure for comparisons by sex and in another figure for comparisons by player position. Use two panels, one for each exercise. It would be appropriate that in addition to the average value, the individual values are presented in figures.
Discussion
Lines 165-166. Sorry, but could you tell me what the term "power-strength characteristics" refers to?
Like the introduction, the authors must present the information in a logical order. That is, first the results regarding sex, second the results regarding the player’s position. For this, the authors should consider the body of literature available on this topic. In addition, much information is lost to explain the difference between certain variables or between the two exercises.
Author Response

(The authors gave the same response as above.)

Reviewer 3 Report
General comments
When using abbreviations, make sure you explain it first and then use the correct abbreviation throughout the paper (i.e. Line 53: “PSC” was already abbreviated).
Introduction
The authors should keep in mind that this section is a development of the hypotheses of the study leading to the purpose of the investigation. At present I think that this section should be improved. Some paragraphs lack for flow, by impacting overall readability of the section.
Furthermore, the authors state in the aim of the study that the paper is also intended to analyze sex differences in rugby players. However, this specific point is not introduced in the introduction section. I would suggest the authors to include a small paragraph related to sex differences in power and strength in rugby players.
Line 53-55: I would specify the level of your athletes (as you mentioned in the title).
Materials and Methods
Participants
Line 62-65: I would highly encourage the authors in keeping consistent the words used in the paper. At this point of the manuscript the reader might not have understood yet the sample of the study. Are they amateur or semi-professional? Furthermore, please provide some reference used in order to classify players level?
Please modify “Kg” in “kg”.
Line 69-70: Do you also have some other measures of weekly volume such as TRIMPs, session RPE, etc.?
Procedures
Line 88-91: How was the velocity controlled? Was it perceived velocity or did you use a metronome?
Maximal Strength
Line 98-100: I am a bit confused about this sentence. Is it part of the warm-up or the warm-up itself? Is it a totally different testing session? Please, provide more info and clarify.
Line 107-109: Were the auditory and visual feedback selected at priori in terms of tempo? How long was the concentric phase?
Power Strength
Which were the indications given to the players?
What about the barbell during the power tests? Was the barbell pushed off? More info about the overall procedures should be provided. Several details such as the barbell push off might influence the power data.
Statistical analysis
Why did not you normalize the strength and power data for body weight in order to compare sex? Please provide some rationales.
Were the power results averaged between the 5 reps for each 30-40-50-60-70-80% 1-RM? How did you process the power data?
Results
You might consider to reduce the decimals in order to make the table 2 more readable and fluent.
I would suggest the authors to not duplicate the results in the text if you already presented them in tables.
Please change “P” with “r”. The “P” is misleading and it is not the proper format to represent Pearson correlation.
Which was the procedure for the correlation analysis? Did you correlate all the measures in your dataset by using all data points or using only the subjects as data point? Please provide more info.
Furthermore, how did you control the collinearity of the variables? There might be the risk in Type 1 error when correlating variables of the same family and the same source of error.
Discussion
Although the results of the study are well discussed with respect to the current literature, I think that some paragraphs lack for flow. I would suggest the authors to reorganize the overall discussion section by keeping the focus on your main results.
Please provide also some practical implications and applications of your results, with specific example.
Line 163-164: Please check the level of your players. They now became national level players.
Author Response

(The authors gave the same response as above.)

Round 2
Reviewer 1 Report
Dear authors, thank you for your reply.
I am aware of the amount of work carried out to come up with data to draft a manuscript and then submit it. The author's expectations are high, but so are the readership expectations.
In this way, my major concern is still the same and was previously addressed: "If there are very large differences between males and females, how can we interpret the differences between backs and forwards when the groups have males and females together?"
The authors replied "... the agility and speed component is greater than in the forwards, it seems logical to think that their PSC is different, especially in the speed parameters where the skills pass speed must be higher"
Well, as you can see the only difference found was in BP max speed. Curiously forwards were faster than Backs (check L 54 of introduction 'Backs are reportedly faster and more agile than forwards'). Probably you would reply that this result was in the BP and running speed derives from the Lower Limbs. Although not significant, in the SQ, forwards were also faster than backs.
You refer in the introduction (L 58) that studies addressing these variables in females are limited. Why don't you explore this gap and explore the female data separately? two groups of 10 and 7 females seems an adequate number of participants to a pioneer study.
When I addressed to normalize data to body mass was to give an idea in relative terms of each test. As I can observe, when absolute strength is required, men are always stronger than females. But when we consider variables that are directly related to personal characteristics, not depending on absolute strength values, we find no differences between males and females (time to max power; time to max speed; SQ max power 1RM%). Of these, only in BP max power 1RM% females got a significantly higher percentage of 1RM to develop max power. It would be worth noting to moment in this result.
Other results, despite not being statistically significant, are those where the backs have higher results than the stronger forwards.
Check with editorial if table on could be placed elsewhere (participants section perhaps).
In L 231 authors refer to a difference between backs and forwards. Well, differences were found (borderline p=.04; ES=0.10). I think you were expecting differences in all variables. Again, females and males backs and forwards were analysed together?
In L240 authors state "The sport position has an influence in the performance showing differences between forwards and backs performance only in maximal speed in BP exercise". If there is only that difference among all variables between backs and forwards (max speed BP), what is really the position's influence?
In L 254 authors state "Apparently, based on our data, it seems that high power levels and the variable speed and time to reach maximum speed can be decisive in performance". Which data/result supports this finding? Power variables (SQ and BP) are different between males and females but no different between backs and forwards; Speed (SQ) are no different between backs and forwards, and time to max speed is no different in any comparison made. There was no specific performance rugby test, as referred to in limitations, to make a statement like this one. Please consider this issue.
Author Response
Dear reviewer,
Thank you for your review and all the comments. We have taken all of them into consideration to improve the quality of the manuscript.
All the best

Reviewer 2 Report
I would like to congratulate the authors for the work done in reviewing the manuscript. However, the structure of the information in short paragraphs/sentences does not facilitate understanding. In this sense, I recommend establishing a series of paragraphs to structure the introduction and the discussion. In addition, some aspects of the materials and methods section still need to be clarified. Finally, the English of the manuscript still needs to improve considerably.
Introduction
Sorry, but the information presented in paragraphs 3 and 4 is redundant with the information presented in paragraphs 1 and 2. Therefore, I recommend that the authors present all the information in 4 paragraphs: 1. contextualization of the demands of rugby, 2) exposition of the importance of strength and power to meet the demands of rugby, 3) review of existing literature comparing strength and power levels based on position and gender, and 4) objectives and hypotheses.
Material and methods
Lines 131-139. I'm sorry, but it's not clear to me how the 1-RM was determined. How could the load of 90%1RM be increased by 20% in the back-squat exercise? In any case, are the four relative loads part of the specific warm-up in addition to the general warm-up (lines 114-117)? This information should be clarified.
Line 155. The linear velocity transducer (T-Force System) measures velocity data instead of power with a sampling frequency of 1,000 Hz.
Lines 157-159. This information is part of the “squat and bench press technique” and not of the “measurement equipment and data analysis”. The authors should explain how the T-Force System calculates the different dependent variables of the present study (e.g. power). In addition, they must inform if the T-Force System is a valid and reliable device in relation to the literature.
Results
Does the table provide any relevant information that is not presented in the text (lines 89-94)?
What information do the * in Table 2 provide? Please consider deleting them.
Provide general results in the text and do not simply reference the tables.
For a better understanding, I recommend that the authors report descriptive values in one figure for comparisons by sex and in another figure for comparisons by player position. Use two panels, one for each exercise. It would be appropriate that in addition to the average value, the individual values are presented in figures.
Discussion
I recommend that the authors group all the information related to each objective in the same paragraph. The fact of dividing the information into small paragraphs makes it difficult for the reader to understand (for example, paragraphs 1 and 2 or paragraphs 3 and 4). In short, the structure could be simpler and clearer in 5 paragraphs: 1) general results, 2) sex, 3) position, 4) contribution, and 5) limitations.
Author Response

(The authors gave the same response as above.)

Reviewer 3 Report
I thank the authors for addressing all my comments. I have no more comments.
Author Response
Dear reviewer,
Thank you for your review and all the comments.
All the best